# The Effectiveness of Free Face Mask Distribution on Use of Face Masks. A Cluster Randomised Trial in Stovner District of Oslo, Norway

**DOI:** 10.3390/ijerph18178971

**Published:** 2021-08-26

**Authors:** Atle Fretheim, Ingeborg Hess Elgersma, Fredrik Aaeng Kristiansen, Caroline Rømming Varmbo, Miriam Kristine Salame Olsbø, Iselin Havstein Strand Glover, Martin Flatø

**Affiliations:** 1Centre for Epidemic Interventions Research, Norwegian Institute of Public Health, 0213 Oslo, Norway; 2Health Services Division, Norwegian Institute of Public Health, 0213 Oslo, Norway; IngeborgHess.Elgersma@fhi.no; 3Research Administrative Support, Norwegian Institute of Public Health, 0213 Oslo, Norway; fredrikaaeng.kristiansen@fhi.no; 4Stovner District-Administration, 0985 Oslo, Norway; caroline.romming@bsr.oslo.kommune.no (C.R.V.); miriam.olsbo@bsr.oslo.kommune.no (M.K.S.O.); iselin.glover@bsr.oslo.kommune.no (I.H.S.G.); 5Centre for Fertility and Health, Norwegian Institute of Public Health, 0213 Oslo, Norway; martin.flato@fhi.no

**Keywords:** face masks, COVID-19, infection control

## Abstract

Face masks are recommended as a means of reducing the spread of COVID-19, but there are practically no studies of interventions to increase face mask use. Over three weeks, nine grocery stores in the Stovner District of Oslo were randomly selected each day to have distribution of free face masks outside their entrance. Free face mask distribution increased the proportion of customers wearing a mask by 6.0 percentage points (adjusted, 95% CI 3.5–8.5). Mean mask usage was 91.7% in the control group and 97.1% in the treatment group (pooled SD 5.3%). Practically all those who wore masks had both nose and mouth covered. We conclude that free distribution of face masks increased their use. Similar trials can be conducted within a short period of time.

## 1. Introduction

In most countries, face masks are recommended as a means of reducing the spread of COVID-19, especially in settings where the risk of transmission is perceived as high, in line with guidance from the World Health Organisation [1]. While many countries and regions have mandated the use of face masks, others have issued recommendations and relied on the public’s voluntary use of masks [2].

Randomised trials of face mask use in the community during the flu season indicate a small protective effect [3]. Only one trial conducted during the COVID-19 pandemic has reported findings so far [4]. While the results were in line with those from earlier trials, the effect estimate was highly uncertain due to few detected cases of COVID-19.

In Norway, there has been no tradition for using face masks in the community and face masks were not recommended by the health authorities in the earliest phase of the COVID-19 pandemic. However, this gradually changed and since October 2020, Norway’s largest city, Oslo, has made use of face masks mandatory when social distancing cannot be maintained in stores and restaurants, on public transport and taxis, and in places of worship [5]. Children under 12 years and people who cannot use face masks for medical or other reasons are exempt from the rule.

Despite a widely recognised need to increase the use of face masks, more than a year into the COVID-19 pandemic there is still a dearth of studies assessing interventions to achieve this. The only studies we are aware of are a large cluster randomised study from Bangladesh, and a few studies that have explored how various forms of messaging or making face mask use mandatory influences peoples’ intentions to wear the masks [6,7,8].

One commonsensical approach to encouraging the use of face masks is free distribution, analogous to providing free hand sanitiser at store entrances and in other strategic places. Free face mask distribution has been introduced in various forms, in several jurisdictions [9,10,11,12,13], and it has some scientific backing from studies of making other types of commodities available for free or at a low price, e.g., condoms for youth [14]. In the recently conducted trial in Bangladesh, free distribution of masks was one component of the multifaceted intervention that was assessed [8].

The Stovner District in Oslo is one of the areas of Norway that has been hardest struck by the pandemic, with around 10% of the population having tested positive for SARS-CoV-2 by the end of April 2021 [15]. The local authorities have implemented several measures for infection control, including mask distribution free-of-charge, by engaging youth who serve as “corona hosts” at key locations. One of the main tasks of the corona hosts is to hand out face masks. Since it is uncertain whether this form of face mask distribution has an impact on face mask use, the Norwegian Institute of Public Health teamed up with the Stovner District administration to carry out a randomised trial of their free distribution scheme. As the intervention targeted groups of people, i.e., customers at a store, a cluster randomised trial was our method of choice.

The objective of our study was to assess whether free distribution of face masks outside grocery stores increased the use of face masks among the customers.

## 2. Materials and Methods

### 2.1. Trial Registration

ClinicalTrials.gov Identifier: NCT04866589. The study protocol was posted on the Norwegian Institute of Public Health website before the study commenced (see Study protocol, Appendix A). Our reporting is in accordance with the CONSORT guideline for cluster randomised trials (see CONSORT checklist, Appendix A).

### 2.2. Setting

Nine grocery stores in the Stovner District in Oslo served as study sites (clusters). There had been no, or only sporadic previous distribution of face masks at these stores. We conducted the trial over 3 weeks from 3 May to 21 May 2021, on weekdays between 1600 h and 1800 h. Due to two national holidays, this constituted 13 days. The COVID-19 incidence during the last two weeks of the trial was 228 per 100,000 inhabitants [16].

### 2.3. Intervention

Young adults from the community were recruited by the district administration to serve as so-called corona hosts, standing in pairs to hand out face masks and hand disinfectant for free outside store entrances (see Figure 1). The masks were standard surgical masks for single use.

### 2.4. Randomisation Procedure

For each day the stores were randomised to free distribution or no free distribution of face masks (see Figure 2).

The members of the research team at the Stovner District administration sent a numbered list of participating stores to one team member at the Norwegian Institute of Public Health-team (A.F.). Another member of the institute team (M.F.) carried out 13 separate and independent randomisations, each assigning the intervention to half the included stores, using Stata software. Eight stores were block randomised to ensure equal number of stores in each group, and a separate randomisation was carried out for a random ninth store. One member of the team (A.F.) merged the list of stores and the randomisation list.

We also randomised the allocation of corona hosts by creating a random ordering of the intervention stores for each day. Since the participating corona hosts could vary from day to day, a numbered list of corona host pairs was sent from the Stovner team to the institute team every morning. The Stovner team had then ensured that each pair was appropriately mixed in terms of experience and gender. The list was then merged with the random order list of intervention stores and returned to the Stovner team who managed the corona hosts.

### 2.5. Analysis

The experiment and analysis were carried out according to the registered protocol. We used a linear model with the proportion using face masks in each store–day as outcome, and controlled for grocery store and day. The regression was weighted by the number of observations, making it equivalent to an individual-based analysis. The controls were levied to reduce standard errors, but their inclusion does not influence the size of the estimated coefficient of interest. We clustered standard errors at the grocery store level to adjust for the fact that the same stores participated each day, so that the observations are not independent across days.

### 2.6. Outcome Measurement

Our main outcome of interest was the proportion of people wearing a mask (without distinguishing between correct or incorrect use). We also assessed the proportion of people wearing a face mask correctly (i.e., covering both mouth and nose).

Outcome measurement was by direct observation. Observers were discreetly placed inside or near the store entrances, after the face mask distribution point (see Figure 3). We had observers posted at all stores for the full duration of the trial, i.e., every weekday from 1600 h to 1800 h. They counted the number of individuals entering the store, noting whether they wore a mask correctly or incorrectly (not covering mouth and nose), or no mask at all. Children under 12 years old were not included. The assessment of whether a person was younger than 12 years old was pragmatically based on the observer’s visual judgement.

### 2.7. Sample Size Estimation

Based on informal reports from the Stovner District administration, we assumed that around 80% of those entering the stores would be using a face mask without free mask distribution, and we assumed a standard deviation of ±10% in means. If free distribution would increase the use of face masks to 90%, we estimated that a trial over the planned study period would be enough to, with reasonable certainty, demonstrate a real difference (5% significance level, 80% power). The estimate was based on simulation exercises, taking into account that each randomised draw is out of the same pool of stores, and that half the variation in means would be within stores. We planned to make an assessment around halfway through the trial, to see if we would need to extend the trial due to lack of statistical power. This halfway assessment was carried out instead of obtaining means and standard deviations from a piloting exercise, which was not desired as it would delay the production of study results in the midst of a pandemic. Although face mask use turned out to be higher and with less variation than anticipated, the halfway evaluation showed that there was sufficient variation to ensure statistical power within the planned trial period.

### 2.8. Ethics and Privacy Issues

This research followed ethical guidelines on research in social sciences, established by the Norwegian research ethics committee [17]. All data we collected was anonymous, so no data protection measures were necessary. Accordingly, when personal data are not collected, the requirement for informed consent of participants can be levied when the research does not imply direct contact with the participants, where the data being processed are not particularly sensitive, and where the utility value of the research clearly exceeds any disadvantages for the individuals involved. Our study fulfils those criteria and hence no consent was collected.

Participants were informed about the study through posters at the store exit. The observers could also point customers to the posters if approached. The posters provided information about the purpose of the research, who had funded the project, the nature of the collected information, dissemination plans for the results, and contact details to project management (see Information poster, Appendix A).

## 3. Results

### 3.1. Descriptive Statistics

During the study period we made 21,524 observations of customers. Descriptive statistics for correct, incorrect, and no face mask use across the nine stores is presented in Table 1. The levels of mask usage in the area was high. Without free face mask distribution, the proportion of customers wearing a mask was 91.7%, and we observed very few cases of incorrect use of face masks. There was also fairly little variation across days and stores in face mask usage; the between-cluster standard deviation was 0.053 for the pooled sample, implying that almost all mean levels of face mask usage that were observed were in the 85–100% range. Face mask use at each store on days with and without face mask distribution is presented in Appendix A.

Furthermore, whereas the average customer in the treatment group shopped in a store with 221 observations per day, the average was 281 in the control group. We have therefore controlled for the number of customers in a robustness check (Appendix A). Controlling for the number of customers did not change the results, and there was no additional effect of the number of customers on face mask usage.

### 3.2. Main Results

We present the estimated effect sizes in Table 2. The regression analysis shows that the distribution of face masks increased face mask usage by 6 percentage points (95% CI 3.5–8.5). Similar effects of the distribution were found for correct usage, which increased by 7.2 percentage points. We observed very few cases of incorrect use of face masks.

Interesting variation in effect sizes is evident when differential effects were considered across stores. These store effects were calculated by an interaction model where the treatment was interacted by store, controlling for day and store, and clustering standard errors at the store level, as in the main specification. Figure 4 plots the effect size by level of face mask usage in the control group. We see that in stores where face mask usage was more prevalent, distributing masks had a smaller effect on usage. Within the range where we have data, a 10 percentage points increase in face mask usage was associated with an eight percentage points reduction in effect size. The store with the lowest control group usage had 73% coverage and saw a 20 percentage points increase due to face mask distribution. Although this is an outlier, the pattern of smaller effects with greater coverage is evident across the sampled range.

In this experiment, the host pairs were also randomly assigned to stores. We may thus study how the effect of the face mask distribution varied by host, again controlling for day and store. Since the composition of host pairs was not random, this analysis rests on the assumption that each host had an independent effect on the face mask distribution so that there was no interaction effect of a well-functioning pair. In total, there were 42 hosts involved in the project (see Figure 5). We see that 13 hosts had significantly positive contributions, ranging from 6–18 percentage points. Only eight hosts had negative effect coefficients, and none of the hosts had significantly negative effects.

There were a few instances when face mask distribution took place at stores that had been allocated to not have face mask distribution, due to misunderstandings or mistakes. There were two such incidences on day 1, and one incidence each of days 2, 3, and 6. In one store on day 9, distribution only happened during 30 min of the two-hour observation window. We therefore conducted an additional analysis where we omitted data from these days. The results provided in Appendix A show that this did not impact on the effect estimate.

## 4. Discussion

We conducted a trial to assess whether free face mask distribution increased the use of face masks in grocery stores. We found that the intervention increased face mask use by 6 percentage points, from a control group mean of 92%.

As this was a randomised controlled trial that strictly followed protocol, we are not aware of major potential sources of bias. Data collection was performed by a single observer at each store, and their judgements of customers’ age and positioning in relation to the distribution point for face masks and to the entrance may have influenced our findings. Still, in most stores we used the one and same observer throughout the trial, so variation in the observers’ judgements should be evenly distributed between observations during days with and without face mask distribution. The fact that the observers were aware of whether face masks were handed out, or not, might have influenced some judgements, though we find it hard to believe that this can have had a sizeable impact. Some degree of bias was certainly introduced when face mask distribution deviated from protocol, but as we have showed, this had little or no influence on our overall findings.

It is a limitation with our study that we did not collect any Appendix A that might explain why some people do not use masks. While our findings could be interpreted to mean that a non-trivial fraction of customers forget to bring a mask or do not wish to purchase face masks for their grocery shopping, an alternative explanation is that the handing out of face masks exerts a form of social pressure that makes it difficult not to wear a mask. Face mask distribution might also have created an expectation that there would be free masks available at the store. A qualitative exploration could have improved our understanding of the mechanisms underlying the observed effect. There was no explicit theory underpinning the intervention, which some may see as a limitation. Still, behavioural change theories can be applied in the interpretation of the findings, e.g., the COM-B model, where behaviour is explained in terms of capability, opportunity, and motivation [18]: Making face masks freely available certainly improved the opportunity for changing behaviour, and social pressure may have increased the motivation for such change.

Baseline adherence to mandatory mask usage was high in our population and our understanding is that face masks were both available, affordable, and acceptable for most members of the community. Keeping that in mind, it would be difficult to imagine that an effect could be larger than the increase of 6 percentage points to reach almost full coverage. We do not know whether these results are transferrable to other settings, with different levels of baseline adherence, access to masks, and attitudes to using them. Still, the heterogeneities in effects that we have explored in this study provide some indication of its external validity. A strong negative association was found between control-group mask usage and the effect of the intervention, which suggests an eight percentage point reduction in effect size for every 10% increase in non-intervention usage within the 73–96% range. This means that the impact of the intervention is likely to be larger in a setting with lower face mask usage, and that the efficiency of an intervention can be vastly improved by targeting it to settings with lower levels of face mask usage.

In our study, only 3% of those who wore face masks were judged to be wearing them incorrectly, i.e., not covering both mouth and nose. This is very low compared to other studies that have observed face mask use in the community, where the proportion of users with only partial coverage of mouth and nose have been reported to be several times higher [19,20,21]. We have no good explanation for this striking difference, but it does illustrate that face mask practices probably vary considerably across contexts.

A unique feature of this study is that the double randomisation of stores and hosts allows for a causal interpretation of the effects of individual hosts on face mask usage. We found that the effects of individual hosts are largely overlapping: only 2 of 42 individuals had significantly larger effects than the worst-performing host. This could be due to similarities between the hosts, as all were young members of the community that they were serving. Another explanation could be that the personal contact between the customer and the host had little impact on the intervention, a contact which perhaps was made more difficult as all hosts wore masks themselves. A limitation of this part of the study was that the pairing of hosts was non-random and estimates of individual effects may thus be impacted by pair interactions.

Our findings demonstrate that free distribution may increase face mask use, but the study should be replicated in other settings. This could be conducted with relative ease, e.g., if local health authorities consider introducing free face mask distribution. The effect of masks on spread of SARS-CoV-2 is uncertain, making it difficult to interpret our findings in terms of impact on COVID-19 incidence [1].

We are aware of one other trial of an intervention to increase community use of face masks: researchers in Bangladesh randomised 600 villages and found that their multifaceted intervention increased face mask use from 13 to 29% [8]. Free distribution of masks constituted only one part of the intervention, so it is not possible to infer how much this contributed to the effect.

The Bangladesh trial was far more ambitious than ours, both in terms of size and in the effort put into the preparatory phase and in developing the intervention. Our study was implemented during a pandemic to provide rapid evidence to stakeholders, and exploring barriers to face mask use and a more elaborate intervention design would have strengthened our study. More clarity is needed on whether economic, social, knowledge-based, or cultural barriers are causing an incomplete adherence to face mask regulations. On the other hand, our pragmatic approach allowed us to finalise the trial in a matter of weeks after the idea to carry out the evaluation was proposed.

There has been a conspicuous lack of trials of infection control measures during the COVID-19 pandemic [22,23]. There are probably a range of different explanations for this, but our study demonstrates that it is possible to conduct policy relevant, simple trials within a short time frame. For us, establishing a close collaboration between researchers and local administrators was key to achieving this.

## 5. Conclusions

Free distribution of face masks outside grocery stores increased the use of face masks in Stovner District of Oslo, Norway. Similar studies are needed to assess whether the findings are valid in other settings. Such trials can be conducted in a matter of weeks.

## Figures and Tables

**Figure 1 ijerph-18-08971-f001:**
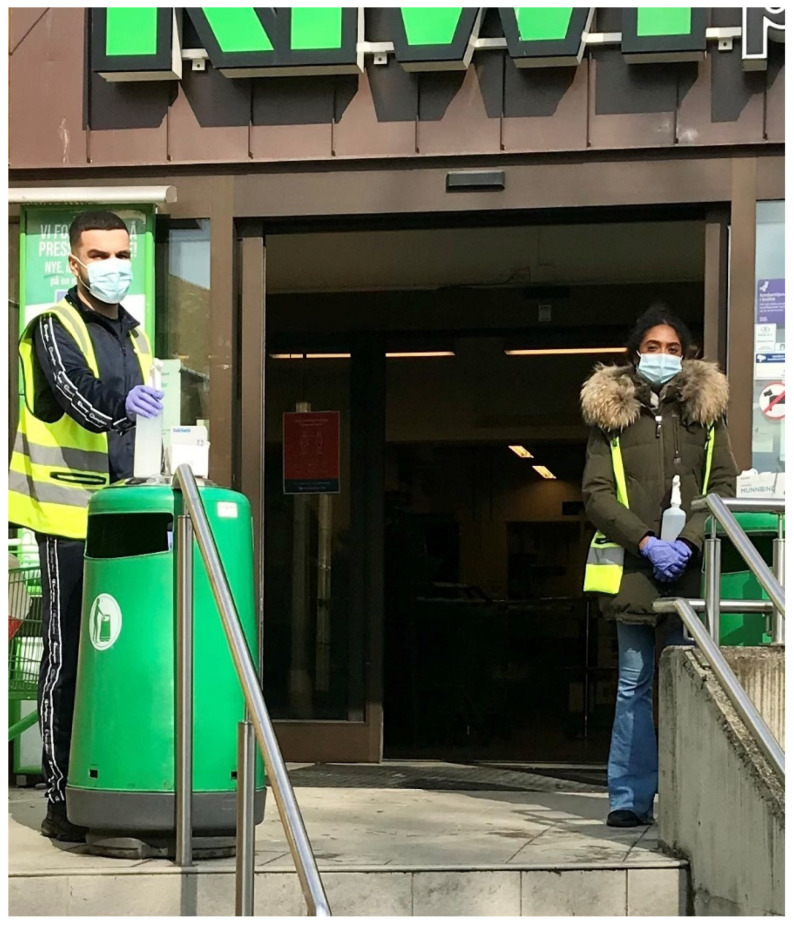
Corona hosts at work.

**Figure 2 ijerph-18-08971-f002:**
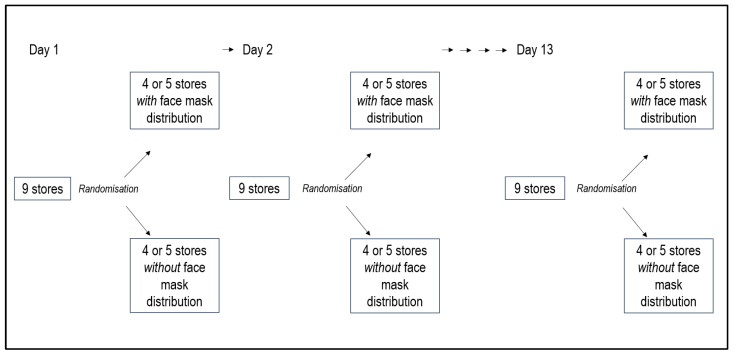
Randomisation procedure.

**Figure 3 ijerph-18-08971-f003:**
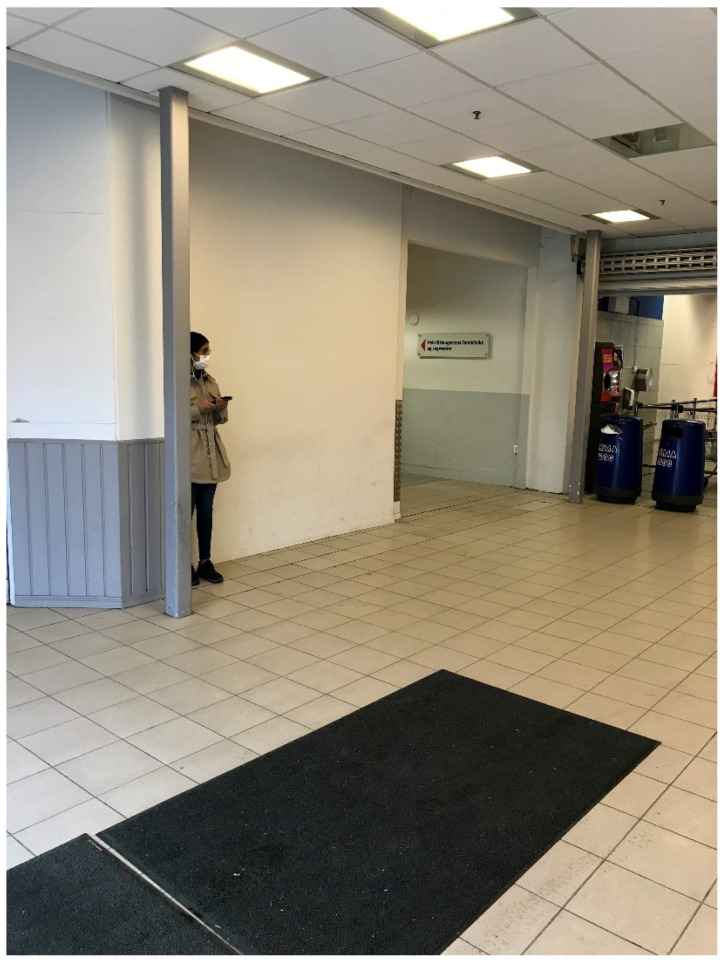
Observer at work.

**Figure 4 ijerph-18-08971-f004:**
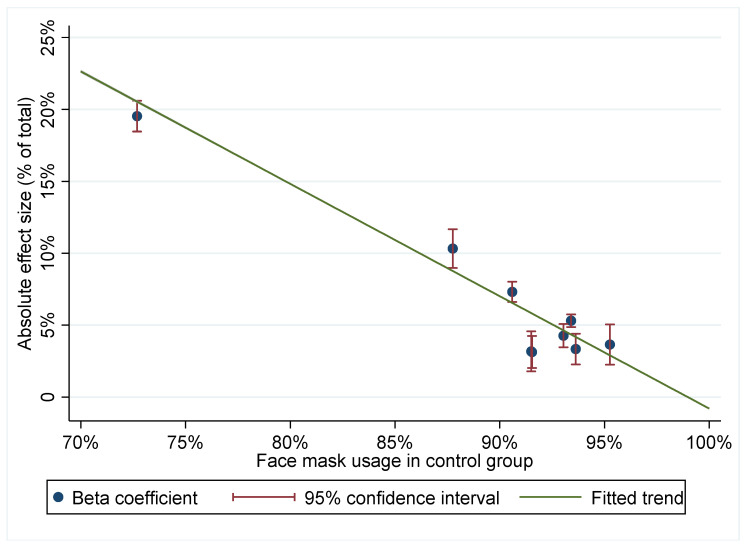
Effect size by level of face mask usage in control group.

**Figure 5 ijerph-18-08971-f005:**
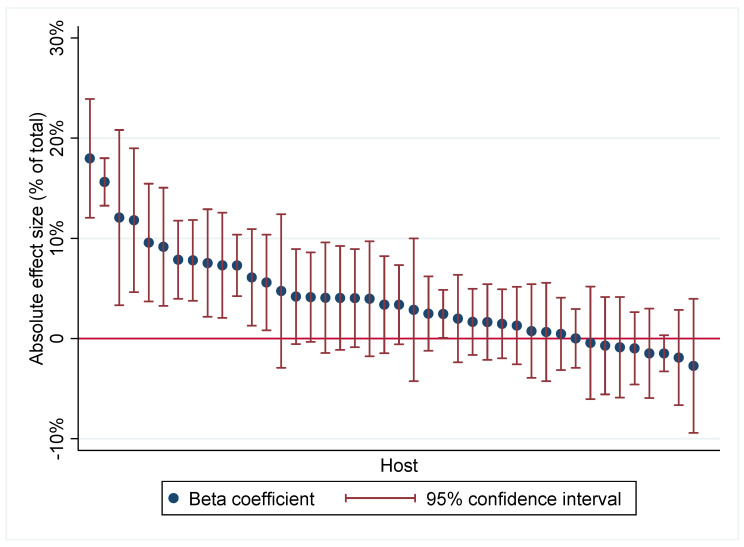
Effect size by host.

**Table 1 ijerph-18-08971-t001:** Descriptive statistics.

	Total	Mask Distribution	No Mask Distribution
Individual level			
Mean (SD) mask usage, percentage	94.4 (23.1)	97.1 (16.6)	91.7 (27.5)
Mean (SD) correct usage, percentage	91.6 (27.8)	95.0 (21.9)	88.4 (32.0)
No. of observations	21,524	11,126	10,398
Cluster level, weighted			
Mean (SD) mask usage, percentage	94.4 (5.3)	97.1 (2.1)	91.7 (6.0)
Mean (SD) correct usage, percentage	91.6 (6.7)	95.0 (3.1)	88.4 (7.5)
Mean (SD) No. of obs.	252 (114)	221 (99)	281 (120)
No. of clusters	117	61	56

Abbreviation: SD, standard deviation.

**Table 2 ijerph-18-08971-t002:** Main results.

	With Face Mask Distribution n/N (%)	Without Face Mask Distribution n/N (%)	Intra-Cluster Correlation Coefficient	Absolute Difference (95% Confidence Interval ^a^)	Relative Difference (95% Confidence Interval ^a^)	*p*-Value ^a^
Face mask use (correct and incorrect)	10,102/10,398 (97.2%)	10,207/11,126 (91.7%)	0.048	0.060 (0.035–0.085)	1.062 (1.036–1.088)	0.001
Correct use of face mask	9873/10,398 (95.0%)	9840/11,126 (88.4%)	0.054	0.072 (0.046–0.099)	1.075 (1.047–1.104)	<0.001

^a^ Adjusted for store and day. Standard errors clustered at the store level.

## Data Availability

The data presented in this article are provided in Appendix A.

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
