# Peer review of "The Effectiveness of Free Face Mask Distribution on Use of Face Masks. A Cluster Randomised Trial in Stovner District of Oslo, Norway"

_ijerph, 2021, doi:10.3390/ijerph18178971_

Round 1
Reviewer 1 Report
This is an intersting study on the role of free face-mask distribution in a grocery setting in achieving higher masking in general population. I have some minor/moderate comments.
"We also collected data on the number of face masks handed out during the study period.] " - why is this in brackets? Is it a leftover from draft version?
"and hence no consent was be collected" - this requires correction
"We see that levels of mask usage in the area was high.", "We furthermore see" - change it to past tense and other verb than "see".
I advise to methodically check the linguistic aspects of the paper to ensure the best quality.
"(Table S2??)." - I presume that you leave the Editor to decide whether this stays?
" There are probably a range of different explanations for this, but our study demonstrates that it is possible to conduct policy relevant, simple trials within a short time frame, at low cost. " - you concluded with focusing on the low cost of intervention. Could you get into some details?
"We also assessed the proportion of people wearing a face mask correctly" - these aspects, although presented in the Results section, are not discussed further. There are several papers on this topic which have been recently executed. This is relevant to discuss (at least briefly) because the dichotomous aspect of wearing vs. not wearing a mask does not guarantee proper adherence to face mask wearing guidelines. So if you wear face masks in a wrong manner there is no additional value in this practice and the whole intervention could be worthless (obviously, not in the whole study group).
Did you collect the data on gender of the participants? Some studies point that women are more prone to comply with safety recommendations.
I like the idea of a "Corona Host" and that you have provided photographic documentation of their work in action. Still, in future studies/interventions I would consider modifying the name to "Anti-Corona Host", as it sounds even more serious and emphasizes the active attitude. "Corona Host" might imply that you work together with the virus, not against it.
Author Response
REVIEWER 1
This is an intersting study on the role of free face-mask distribution in a grocery setting in achieving higher masking in general population. I have some minor/moderate comments.
"We also collected data on the number of face masks handed out during the study period.] " - why is this in brackets? Is it a leftover from draft version?
RESPONSE: Yes, apologies from our side. This is simply a “leftover”.
"and hence no consent was be collected" - this requires correction
REPONSE: Thanks, corrected! (By deleting “be”).
"We see that levels of mask usage in the area was high.", "We furthermore see" - change it to past tense and other verb than "see".
RESPONSE: We have changed accordingly: “The levels of mask usage in the area.”, “Furthermore, whereas”…
I advise to methodically check the linguistic aspects of the paper to ensure the best quality.
RESPONSE: We have reviewed the text and made a series of adjustments (improvements), languagewise.
"(Table S2??)." - I presume that you leave the Editor to decide whether this stays?
RESPONSE: Apologies again – this is another “leftover” from an earlier draft. Corrected.
" There are probably a range of different explanations for this, but our study demonstrates that it is possible to conduct policy relevant, simple trials within a short time frame, at low cost. " - you concluded with focusing on the low cost of intervention. Could you get into some details?
RESPONSE: When we point at the “low cost”, we mean the cost of conducting the evaluation – not the cost of the intervention. We have deleted “at low cost”, to avoid any confusion.
"We also assessed the proportion of people wearing a face mask correctly" - these aspects, although presented in the Results section, are not discussed further. There are several papers on this topic which have been recently executed. This is relevant to discuss (at least briefly) because the dichotomous aspect of wearing vs. not wearing a mask does not guarantee proper adherence to face mask wearing guidelines. So if you wear face masks in a wrong manner there is no additional value in this practice and the whole intervention could be worthless (obviously, not in the whole study group).
RESPONSE: We agree that it makes sense to comment on the (very low) incidence of incorrect use in our study, compared to what others have reported based on similar methods, so we have added a short paragraph addressing this, in the Discussion-section.
Did you collect the data on gender of the participants? Some studies point that women are more prone to comply with safety recommendations.
RESPONSE: No, we did not collect data on gender.
I like the idea of a "Corona Host" and that you have provided photographic documentation of their work in action. Still, in future studies/interventions I would consider modifying the name to "Anti-Corona Host", as it sounds even more serious and emphasizes the active attitude. "Corona Host" might imply that you work together with the virus, not against it
RESPONSE: We agree fully! We must admit that the term “Corona host” (in Norwegian, “koronavert”) had become so familiar to us that had not thought about this unfortunate (and unintentionally comical) meaning of the term.
Reviewer 2 Report
I appreciate the opportunity to review this manuscript. The purpose of this study is to assess whether free distribution of face masks outside grocery stores increased the use of face masks among the customers. It was an interesting manuscript in the context of recent concerns about the fourth infection caused by the Delta variant.
I’m going to start by outlining three major concerns that pertain to the design and cannot be remedied within the confines of the existing study. Unfortunately, three of the concerns are of an extent to which I’m left with very little confidence in the study’s conclusions.
The first reason is that in reality, if countries that distribute masks for free seem to be rare, the author should consider explaining why it is necessary to study a causal relationship. I wish there was an explanation as to which countries around the world are distributing masks for free. However, in reality, few countries are distributing masks for free due to the lack of government budget. If we distribute masks for free, won't the mask usage rate increase automatically? The necessity of this study is not well explained.
Second, little focus is put on the theoretical contribution of the authors’ results, linking these results back to the existing literature. This is vital.
Lastly, has this study passed the IRB? If this study falls under IRB exemption, it must be approved by the IRB that it falls under IRB exemption.
Hope my comments are helpful.
Author Response
REVIEWER 2
I appreciate the opportunity to review this manuscript. The purpose of this study is to assess whether free distribution of face masks outside grocery stores increased the use of face masks among the customers. It was an interesting manuscript in the context of recent concerns about the fourth infection caused by the Delta variant.
I’m going to start by outlining three major concerns that pertain to the design and cannot be remedied within the confines of the existing study. Unfortunately, three of the concerns are of an extent to which I’m left with very little confidence in the study’s conclusions.
The first reason is that in reality, if countries that distribute masks for free seem to be rare, the author should consider explaining why it is necessary to study a causal relationship. I wish there was an explanation as to which countries around the world are distributing masks for free. However, in reality, few countries are distributing masks for free due to the lack of government budget. If we distribute masks for free, won't the mask usage rate increase automatically? The necessity of this study is not well explained.
RESPONSE: We agree that it seems plausible that free distribution of face masks should lead to increased use of face masks. However, whether this is the case in practice is yet to be shown, and for decision makers who may be considering whether to implement a free distribution scheme, the effect size is certainly of importance. In fact, our study was initiated for that exact reason: The health authorities in the area (Stovner in Oslo) had already implemented free distribution of face masks in some locations, but where unsure of whether the impact was worth the effort. We disagree with the claim the distribution of masks for free is rare – at least in Norway free face mask distribution is quite commonplace, e.g. in shopping centres. Also, health authorities in other countries have introduced various forms of free face mask distribution,e.g.:
- https://www.bristolpost.co.uk/news/bristol-news/80000-face-masks-being-handed-4356307
- https://www.cnet.com/health/who-are-bidens-25-million-free-covid-19-face-masks-for-what-we-know-so-far/ .
Furthermore, this intervention was conducted in a neighbourhood that had been struck particularly hard by the pandemic. Evidence of impact in such strategic locations may stimulate similar efforts elsewhere that do not need to expand into nation-wide permanent roll-outs at high cost.
Second, little focus is put on the theoretical contribution of the authors’ results, linking these results back to the existing literature. This is vital.
RESPONSE: As there have been practically no other studies of the effect of interventions to increase use of face masks, there is little existing literature to refer to. We have compared our findings with another recent trial, in the Discussion-section. In addition, we have now added a short paragraph in the Discussion where we refer to 3 previous studies of face mask using behaviour.
Lastly, has this study passed the IRB? If this study falls under IRB exemption, it must be approved by the IRB that it falls under IRB exemption.
RESPONSE: We have not applied for IRB-approval, as this type of study falls outside the mandate of Norwegian IRBs (Ethical review committees). As described in the Ethics and privacy issues-section of the manuscript, we have adhered to the current guidelines for ethical conduct of these types of studies.
Hope my comments are helpful.
Reviewer 3 Report
Fretheim et al. present results of a 13-day cluster randomised trial in Oslo assessing whether free face mask distribution has an effect on use of face masks. Data given are not completely novel as the authors state themselves in the discussion section, but can be interesting for stakeholders and policy in the current pandemic. The manuscript is well written an easy to understand. I only have some minor points, that could be addressed:
1) Introduction: The authors could provide an additional paragraph detailing with the common political discussion of free distribution of face masks.
2) Table 1 and 2: How is it possible that the distribution influences the correct use of face masks. Please state on this.
3) "Figure 1" is stated twice. Please change one to "Figure 2".
4) Figure 4: The gray bar "95% CI on trend" cannot be found in the figure itself.
5) Discussion, line 223: a randomised controlled trial can only control for confounders not for bias. Please change "bias" to "confounders".
6) Discussion: This could be phrased a bit more concisely, concentrating on the interpretation of mentioned results and listing the limitations of the study at the end.
Author Response
REVIEWER 3
Fretheim et al. present results of a 13-day cluster randomised trial in Oslo assessing whether free face mask distribution has an effect on use of face masks. Data given are not completely novel as the authors state themselves in the discussion section, but can be interesting for stakeholders and policy in the current pandemic. The manuscript is well written an easy to understand. I only have some minor points, that could be addressed:
RESPONSE: Thank you!
- Introduction: The authors could provide an additional paragraph detailing with the common political discussion of free distribution of face masks.
RESPONSE: Free distribution has been implemented in various ways in some countries, and we have now added a sentence about this, including references to news-reports on the issue. However, we are not aware of extensive political debates on the issue, and we have not been able to identify sources describing in any detail.
- Table 1 and 2: How is it possible that the distribution influences the correct use of face masks. Please state on this.
RESPONSE: We don’t have a clear understanding or explanation as to why face mask distribution increased the use of face masks, including correct use of masks. However, the proportion of face mask users in the intervention and control groups were not very different (97.7% and 96.4%), so we don’t think there is good reason to believe that free distribution increased correct use as such. It seems more reasonable to believe that the general increase in use of face masks was accompanied by a corresponding increase in correct use of masks.
- "Figure 1" is stated twice. Please change one to "Figure 2".
RESPONSE: Thank you for noticing – corrected!
- Figure 4: The gray bar "95% CI on trend" cannot be found in the figure itself.
RESPONSE: We have removed the grey areas from the legend.
- Discussion, line 223: a randomised controlled trial can only control for confounders not for bias. Please change "bias" to "confounders".
RESPONSE: The point we made was that the combination of this being a) “a randomised controlled trial”… and b) that it “strictly followed protocol”, means that we can reasonably rule out major factors which might have led to biased findings, including i) allocation bias (due to problems with the randomization process); ii) deviations from the assigned intervention/control or lack of adherence to assigned intervention/control; iii) missing outcome data; iv) outcome measurement; or v)outcome reporting. We are aware that there are different terminologies in this area, but we feel comfortable leaning on the understanding of “risk of bias”-assessment that the Cochrane Collaboration and others use. However, we are willing to adjust the wording, if asked to by the editors. It would be misleading to state that an RCT controls for confounders. The identification strategy does not rely on adding control variables to adjust for observable characteristics which the phrase “controlling for” would allude to, but on randomisation that removes potential bias that might otherwise be present due to both observable and unobservable differences between the control and intervention groups.
6) Discussion: This could be phrased a bit more concisely, concentrating on the interpretation of mentioned results and listing the limitations of the study at the end.
RESPONSE: Extending from the previous point, the main limitation of our study – the way we see it – is that the findings cannot be assumed to be applicable in other settings. Since we address this explicitly in the Conclusion, we feel that we are indeed pointing out the study’s main limitation in the conclusion, already. We have revised the paragraph about the Bangladesh, study, to make it more consise.
Round 2
Reviewer 2 Report
Thank you for the opportunity to review this manuscript. The purpose of this study was to assess whether free distribution of face masks out-66 side grocery stores increased the use of face masks among the customers. This study has the advantage that it was conducted through a rigorous procedure. The following are my comments about this research.
First, there is no such mention in revision despite my comment asking if there is a country that distributes masks for free. As I mentioned in the first review, it is difficult to empathize with the necessity of this study.
Second, there is no mention of the theoretical contribution of this study. The reviewer believes that IJERPH as an academic journal publishes manuscripts with a theoretical analysis framework.
Third, the reviewer doesn't know why the authors wrote the following sentence in the abstract:
“Similar trials can be conducted within a short period of time, at low cost.”
Lastly, the reviewer believes that the authors should consider omitting “the” in the following sentence (p.7): “We see that the in stores where face mask usage 194 was more prevalent, distributing masks had a smaller effect on usage.”
Unfortunately, it seems difficult to recommend publishing this manuscript in IJERPH. Hope my comments are helpful.
Author Response
Reviewer 2 comments, second round, with authors' response.
Thank you for the opportunity to review this manuscript. The purpose of this study was to assess whether free distribution of face masks out-66 side grocery stores increased the use of face masks among the customers. This study has the advantage that it was conducted through a rigorous procedure. The following are my comments about this research.
First, there is no such mention in revision despite my comment asking if there is a country that distributes masks for free. As I mentioned in the first review, it is difficult to empathize with the necessity of this study.
RESPONSE:
We may not have been clear enough when we formulated our response, but we did indeed make such a change: We added references to free face mask distribution in other countries when we responded to this comment in the previous round:
One common-sensical approach to encouraging the use of face masks is free distribution. This strategy has been introduced in various forms, in several jurisdictions [9-13]
(The text in bold and references 9 to 13 were added in direct response to the reviewer’s comment, in the previous round).
***************************
Second, there is no mention of the theoretical contribution of this study. The reviewer believes that IJERPH as an academic journal publishes manuscripts with a theoretical analysis framework.
REPONSE:
We have the same response as we have to the similar comment from the academic editor:
Our study was a purely pragmatic evaluation of an existing intervention (free face mask distribution) that had already been partly implemented, and we do not see significant value in trying to fit it into a theoretical framework. However, recognizing that one reviewer and the academic editor see this differently, we are willing to add the following in the Discussion-section:
There was no explicit theory underpinning the intervention, which some may see as a limitation. Still, behavioural change theories can be applied in the interpretation of the findings, e.g. the COM-B model, where behaviour is explained in terms of capability, opportunity, and motivation [18]. Making face masks freely available certainly improved the opportunity for behaviour change, and social pressure may have increased the motivation for change.
***************************
Third, the reviewer doesn't know why the authors wrote the following sentence in the abstract:
“Similar trials can be conducted within a short period of time, at low cost.”
RESPONSE:
We explain clearly in the full manuscript that the study took place within a short timeframe. We disagree that this explanation is needed in the abstract since there is no space for this, and we believe the reader will look at the full text if clarification or further explanation is needed after reading the abstract. However, we realise that we have not reported much on resource use. In response we have therefore deleted the terms “low cost” and “limited resources”.
***************************
Lastly, the reviewer believes that the authors should consider omitting “the” in the following sentence (p.7): “We see that the in stores where face mask usage 194 was more prevalent, distributing masks had a smaller effect on usage.”
RESPONSE:
Thanks, changed accordingly.
